# Linsitinib inhibits IGF-1-induced cell proliferation and hyaluronic acid secretion by suppressing PI3K/Akt and ERK pathway in orbital fibroblasts from patients with thyroid-associated ophthalmopathy

Ji-Young Lee[1], Seong-Beom Lee[1], Suk-Woo Yang[2], Ji-Sun Paik[3]*

1 Department of Pathology, Institute of Hansen's Disease, College of Medicine, The Catholic University of Korea, Seoul, Korea, 2 Department of Ophthalmology and Visual Science, Seoul St. Mary's Hospital, College of Medicine, The Catholic University of Korea, Seoul, Korea, 3 Department of Ophthalmology and Visual Science, Yeouido St. Mary's Hospital, College of Medicine, The Catholic University of Korea, Seoul, Korea

* rollipopp@daum.net

**Data Availability Statement:** Data are all contained within the paper and/or Supporting information files.

## Abstract

Thyroid-associated ophthalmopathy (TAO), an autoimmune disorder of the retrobulbar tissue, is present in up to 50 percent of Graves's hyperthyroidism patients. Insulin-like growth factor 1 receptor (IGF-1R) has received attention as a target for the development of therapeutic agent for TAO. IGF-1R and TSHR (thyroid stimulating hormone receptor) interact with each other to form a physical or functional complex, further promoting the development of TAO. Linsitinib, OSI-906, is an inhibitor of IGF-1R and has been reported to inhibit cell proliferation of several tumor cells. Linsitinib has been receiving attention not only for its anti-cancer effect, but also for its anti-inflammatory effects. It has been reported that linsitinib reduces infiltration of inflammatory cells in orbital tissues, resulting in the reduction of muscle edema and adipose tissues in an experimental murine model for Graves' disease. In the current study, we investigated the issue of whether linsitinib inhibits the IGF-1-induced proliferation of orbital fibroblasts (OFs) via the suppression of phosphatidylinositol 3-kinase (PI3K) / Akt and extracellular signal-regulated kinase (ERK) pathway. Our results showed that pretreatment with linsitinib inhibited IGF-1-induced cell proliferation and hyaluronic acid secretion in the OFs of TAO patients. In addition, our results showed that pretreatment with linsitinib inhibited IGF-1-induced phosphorylation of IGF-1Rβ at Tyr1135, Akt at Ser473, and ERK in the OFs of patients with TAO. These results indicate that linsitinib inhibits IGF-1-induced cell proliferation and hyaluronic acid secretion in the OFs of TAO patients by suppressing the PI3K/Akt and ERK pathways, validating the use of linsitinib as a novel therapeutic agent for TAO.

**Funding:** This work supported by the Department of Ophthalmology, College of medicine, the Catholic University of Korea, Ki-Soo Kim Research Fund, 2023.

**Competing interests:** No authors have competing interests.

## Introduction

Thyroid-associated ophthalmopathy (TAO) is an autoimmune disorder of the retrobulbar tissue most commonly associated with hyperthyroid or euthyroid. Ophthalmic manifestations are present in up to 50 percent of Graves's hyperthyroidism patients [1].

Thyroid-stimulating immunoglobulin (TSI) binds to the thyroid-stimulating hormone receptor (TSHR) in orbital tissues, and the resulting immune response leads to periorbital edematous and inflammatory changes, including the infiltration of inflammatory cells, the accumulation of extracellular matrix (ECM) proteins, the proliferation of fibroblasts and increased levels of fatty tissue, resulting in the development of clinical signs of TAO [2]. The clinical signs include widening of the palpebral fissure, eye lid retraction, lid lag, conjunctival congestion, chemosis, proptosis, corneal exposure, restrictive myopathy and optic neuropathy [3].

IGF-1 potentiates the effects of TSH and TSI in the production of hyaluronic acid and phosphorylation of extracellular signal-regulated kinase (ERK) by orbital fibroblasts (OFs) [4]. The IGF-1 receptor (IGF-1R) has garnered attention as being involved in the development of TAO [5]. Inhibiting IGF-1R with an anti-IGF-1R antibody, teprotumomab, is an effective and safe way of treating moderate to severe TAO [6].

IGF-1R exists in homodimer form, and each monomer consists of an extracellular domain, a transmembrane, and a cytoplasmic tyrosine kinase domain. The corresponding ligand, IGF-I and IGF-II, binding to IGF-1R activates intracellular intrinsic kinase, which induces the autophosphorylation of IGF-1R and subsequently promotes various signal transductions such as phosphatidylinositol 3-kinase (PI3K) /Akt and ERK pathway, leading to cell proliferation and hyaluronic acid secretion [7].

Linsitinib, also known as OSI-906, inhibits the intrinsic kinase activity of IGF-1R by binding to the cytoplasmic kinase domain. Especially, linsitinib inhibits the phosphorylation of IGF-1R and insulin receptor (IR), subsequently resulting in the blockade of IGF-1-induced activation of downstream signaling pathway, PI3K /Akt and ERK pathway [8]. Yeo CD et al. [9] reported that linsitinib suppressed IGF-1-induced cell proliferation and phosphorylation of IGF-1R in various tumor cell lines. In addition, it has been reported that treatment with linsitinib reduces infiltration of T-lymphocytes and macrophages in orbital tissues, resulting in the reduction of muscle edema and adipose tissues in an experimental murine model for Graves' disease [10]. Thus, we thought that linsitinib would also affect the activity of orbital fibroblast.

In the current study, we investigated whether linsitinib inhibits the IGF-1-induced proliferation of OFs and hyaluronic acid (HA) secretion by OFs via the suppression of the PI3K/Akt and ERK pathways.

## Materials and methods

### Reagents and antibodies

Linsitinib was purchased from SelleckChem (Houston, TX). Linsitinib was dissolved in dimethyl sulfoxide (DMSO). The final vehicle concentration was adjusted to 0.1% (v/v), and the control medium contained the same quantity of vehicle. Antibodies against cyclin D1, phospho-Akt (Thr308), phospho-Akt (Ser473), Akt, phospho-ERK, ERK, phospho-IGF-1Rβ (Tyr1135) and IGF-1Rβ were obtained from Cell Signaling Technology (Beverly, MA). Antibody against GAPDH was obtained from Santa Cruz Biotechnology (Santa Cruz, CA). Horseradish peroxidase-conjugated secondary antibodies were obtained from Merck Millipore (Burlington, MA).

## Cell cultures

Human OFs were obtained from orbital fat from decompression surgery in patients with TAO. Orbital fat explants were minced in small pieces and placed in a T25 culture flask. The culture media was then poured into the culture flask until the floating fat tissues reached the top plate. The culture medium used in the current study was Dulbecco's Modified Eagle's medium (GIBCO BRL, Grand Island, NY) supplemented with 20 mM HEPES (Fisher Scientific, Atlanta, GA), 10% fetal bovine serum (FBS; GIBCO BRL), 100 U/ml of penicillin, and 100 µg/ml of streptomycin (Bio Whittaker Inc., Walkersville, MD). Cultures were maintained at 37°C in a 5% $CO_2$ humidified incubator until the fibroblasts reached 70% confluence at the top plate of the culture flask. Non-adherent cells and fat tissues were then removed, and the established fibroblasts were maintained in a 100 mm culture plate and passaged with gentle trypsin/EDTA treatment. Fibroblasts were not used for studies beyond passage 10 from the initial culture. The control subjects group had no systemic diseases or inflammatory/immune disease. In the control group, the orbital fats were obtained from patients who had upper or lower eyelid blepharoplasties or other orbital surgeries. The Charateristics of individual patients with TAO/ without TAO are described in Table 1. Institutional Review Board/Ethics Committee of Seoul St. Mary's Hospital approved this research (KC10TISE0743). Informed written consent was obtained from the donors prior to OFs isolation, according to guidelines from the Institutional Review Board of Seoul St. Mary's Hospital. The study was conducted in accordance with the tenets of the Declaration of Helsinki.

## Cell proliferation assay

The assay for cell proliferation was performed by using a CCK-8 (Cell Counting Kit-8™, Dojindo Molecular Technologies Inc., Kumamoto, Kyushu, Japan). Human orbital fibroblasts were plated at $5 \times 10^3$ cells/well in a 96-well plate. After 24 h, cells were starved in DMEM containing 0.1% BSA for 24 h. The serum starved cells were pre-treated with 20 µM linsitinib for 2 h, followed by treatment with 50 ng/mL IGF-1 for 24 h. After adding 10 ul of CCK-8 solution to each well, the cells were incubated for 2 h. The optical density for each well was then measured at 450 nm in a spectrometer (µ-Quant, BioTek instruments, Inc.).

**Table 1. Characteristics of patients with thyroid-associated ophthalmopathy (TAO) and subjects without TAO (non TAO) from whom orbital fibroblasts were obtained in this study.**

|  | Patients with TAO | Subjects without TAO (non TAO) |
|---|---|---|
|  | (n = 3) | (n = 2) |
| Mean age, years (range) | 55 (43–67) | 58(51–65) |
| Sex (m/f) | 2/1 | 1/1 |
| Smoking history | 2 | 1 |
| Graves' disease |  |  |
| Radioactive iodine therapy | 0 | - |
| Surgery | 0 | - |
| Methimazole | 3 | - |
| Treatment of TAO |  |  |
| Surgery | 3 | - |
| prednisolone | 1 | - |
| Radiation | 1 | - |
| Euthyroid | 3 | 3 |
| TSH receptor antibodies | 3 | 0 |
| CAS (range) | 2,3,4 | - |

## Cell cycle analysis

The treated OFs were analyzed for DNA content using a Propidium Iodide Flow Cytometry Kit. (Abcam, Cambridge, UK). Human OFs were harvested, washed in PBS, and fixed 2 h in cold 66% ethanol at 4°C. The cells were centrifuged at 3,000 rpm for 5 min and then washed in PBS. Cells were stained with propidium iodide staining solution containing 100 μg/mL RNase A and dark incubated for 30 min at 37°C. DNA contents were analyzed by flow cytometry (FACS Cantro II, BD Bioscience, San Jose, CA).

## Quantification of HA

Human OFs were plated at $2x10^5$ cells/well in a 6-well plate. After 24 h, cells were pre-treated with 1, 5, and 20 μM linsitinib for 2 h, followed by treatment with 100 ng/mL IGF-1 for 72 h. Culture medium was analyzed for HA concentration using a HA-enzyme linked immunosorbent assay (ELISA) kit (*Corgenix*, Westminster, USA) according to the manufacturer's instructions. The optical density for each well was then measured at 450 nm in a spectrometer (μ-Quant, BioTek instruments, Inc.).

## Cell viability assay

Cell viability was evaluated using the 3-(4,5-dimethylthiazol-2-yl)-2,5-diphenyltetrazolium bromide (MTT) reduction assay in 96-well plate. Cells were treated based on the experimental protocols and 10 μl of a 5 mg/ml MTT solution was then added to each well. After incubation in a 5% $CO_2$ incubator for 2 h at 37°C, media were removed and 100 μl HCl-isopropyl alcohol were added to each well. After incubation for 10 min, 100 μl distilled water were added to each well. The optical density for each well was then measured at 570 nm in a spectrometer (μ-Quant, BioTek instruments, Inc., Winooski, VT).

## Western blot analysis

The treated OFs were removed from the incubator at the designated times and placed on ice. The cells were then washed 3 times with ice cold PBS. The cells were then lysed for suspending them for 30 min in RIPA lysis buffer [50 mM Tris–HCl (pH 7.4), 1% Triton X-100, 150 mM NaCl, 0.1% sodium dodecyl sulfate (SDS), 0.5% sodium deoxycholate, 100 mM phenylmethylsulfonyl fluoride, 1 μg/ml of leupeptin, 1 mM Na3VO4, and 1X Complete™ Protease Inhibitor Cocktail (Santa Cruz Biotechnology)]. Equal amounts of protein were loaded onto 8–15% SDS-PAGE gels, electrophoresed, and transferred to nitrocellulose membranes (Bio-Rad Laboratories, Inc. Berkeley, CA). The membranes were blocked in Tris-buffered saline with 0.05% Tween-20 (TBST) supplemented with 5% powdered milk and then incubated with a primary antibody against the designated protein. The blots were then washed with TBST and incubated with a horseradish peroxidase conjugated secondary antibody in TBST plus 5% powdered milk. The bound antibodies were detected with the SuperSignalTM West Dura Extended Duration Substrate (Thermo Fisher Scientific Inc. Waltham, MA).

## Statistical analysis

All results are expressed as the means ± SD of data from at least three separate experiments. Statistical significance was determined via one-way ANOVA. $P < 0.01$ was considered to be statistically significant.

## Results

### Pretreatment with linsitinib inhibits IGF-1-induced proliferation of OFs from TAO patients

We initially assessed the issue of whether linstinib affects IGF-1-induced proliferation of OFs. Treatment with 50 ng/ml IGF-1 significantly increased the proliferation of OFs from TAO patients in DMEM containing 0.1% bovine serum albumin (Fig 1C). However, the stimulatory effect of IGF-1 on cell proliferation was not significant in DMEM containing 1% FBS and 10% FBS (Fig 1C). Thus, in the following experiments, the effect of IGF-1 on cell proliferation was examined using DMEM containing 0.1% BSA. As shown in Fig 1A, pretreatment with 20 μM linsitinib significantly reduced IGF-1-induced proliferation of OFs from TAO patients.

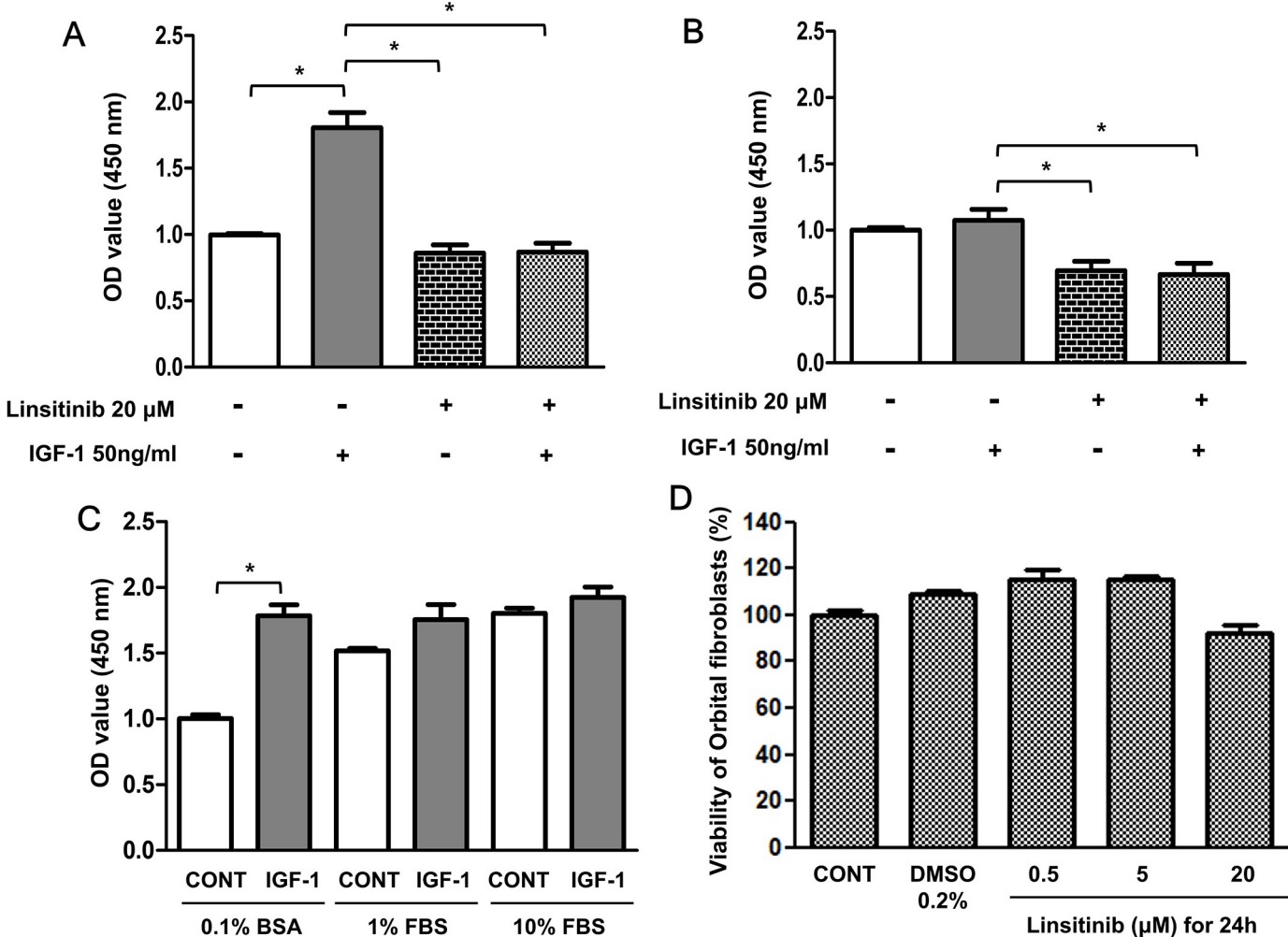

**Fig 1. Pretreatment with linsitinib inhibits IGF-1-induced cell proliferation in OFs from TAO or non-TAO patients.** (A and B) OFs from TAO or non-TAO patients were plated at a concentration of 5 x 10³ cells/well in a 96-well plate. After 24 h, cells were starved in DMEM containing 0.1% BSA for 24 h. The serum starved cells were pre-treated with linsitinib at the indicated concentrations for 2 h, followed by treatment with 50 ng/ml IGF-1 for 24 h. Similar results were observed in three independent experiments. (C) OFs from TAO patients were plated at a concentration of 5 x 10³ cells/well in a 96-well plate. After 24 h, cells were treated with 50 ng/ml IGF-1 in DMEM containing 0.1% BSA, 1% FBS or 10% FBS for 24 h. Cell proliferation was determined using a Cell Counting Kit-8 assay. *$P < 0.01$ between the indicated groups. (D) OFs from TAO patients were treated with linsitinib at the indicated concentrations for 24 h. Cell viability was determined using an MTT reduction assay.

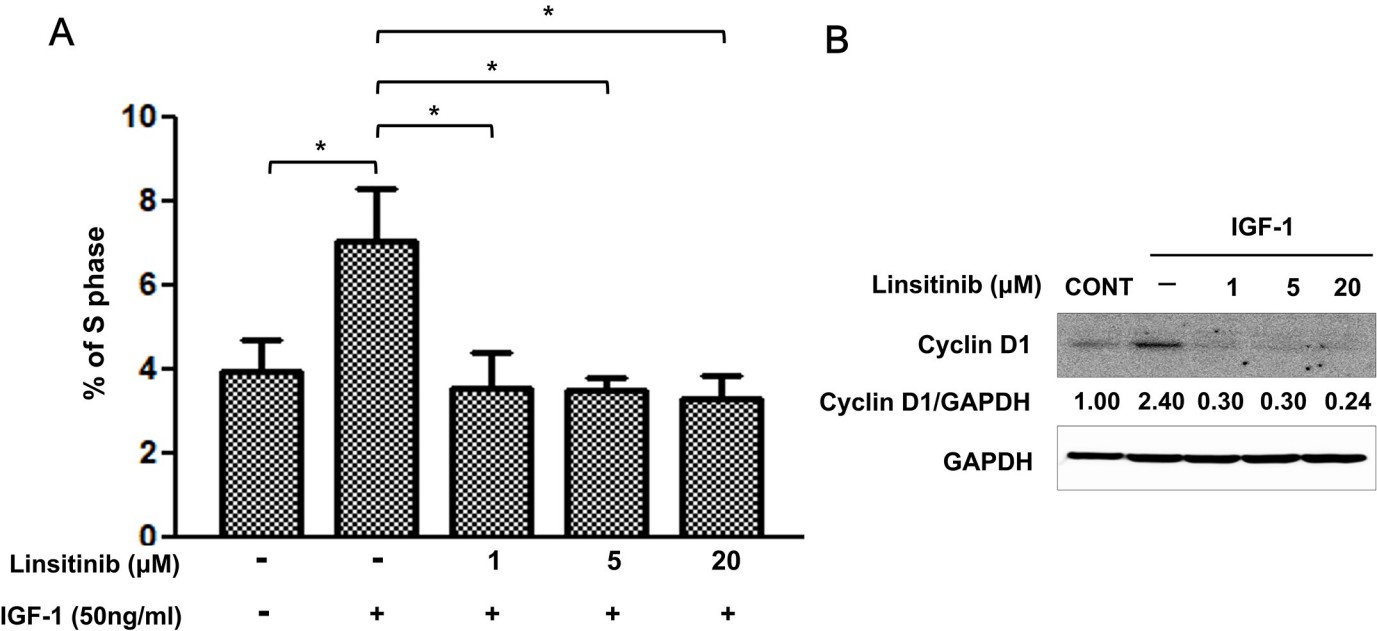

**Fig 2. Pretreatment with linsitinib reduces IGF-1-increased proportion of S-phase cells in OFs from TAO patients.** (A) OFs from TAO patients were plated at a concentration of $2 \times 10^5$ cells/well in a 6-well plate. After 24 h, the cells were pre-treated with linsitinib at the indicated concentrations for 2 h, followed by treatment with 50 ng/ml IGF-1 for 24 h. After 24 h, cells were collected and stained with propidium iodide (PI) and their DNA contents were analyzed by flow cytometry. The percentage of cells in the S phase of the cell cycle. *$P < 0.01$ and **$P < 0.05$ between the indicated groups. Similar results were observed in three independent experiments. (B) OFs from TAO patients were plated at a concentration of $2 \times 10^5$ cells/well in a 6-well plate. After 24 h, cells were starved in serum-free medium for 24 h. The serum starved cells were pre-treated with linsitinib at the indicated concentrations for 2 h, followed by treatment with 50 ng/ml IGF-1 for 5 min. The levels of expression of cyclin D1 were determined via western blot analysis. Similar results were observed in three independent experiments.

However, treatment with 50 ng/ml IGF-1 did not increase the proliferation of OFs from control subjects (Fig 1B). Therefore, in this study, we only studied the effect of linstinib on OFs from TAO patients. Treatment with linsitinib at concentrations up to 20 μM had no effect on viabilities of OFs (Fig 1D).

## Pretreatment with linsitinib reduces IGF-1-increased the proportion of S-phase cells in OFs from TAO patients

We then examined the issue of whether linstinib inhibits IGF-increased the proportion of S-phase cells in OFs from TAO patients. Treatment with 50 ng/ml IGF-1 significantly increased the S-phase proportion of OFs from TAO patients from 3.9% to 7.0% (Figs 2A and S1B). Consistent with the inhibitory effects of linsitinib on cell proliferation (Fig 1A), pretreatment with 1–20 μM linsitinib significantly reduced IGF-increased the proportion of S-phase cells in OFs from TAO patients (Figs 2A and S1D). In addition, pretreatment with 1–20 μM linsitinib inhibited the IGF-1-induced expression of cyclin D1 in OFs from TAO patients (pretreatment with 20 μM linsitinib + treatment with IGF-1 by 0.24 fold vs only IGF-1 treatment by 2.40 fold, Fig 2B).

## Pretreatment with linsitinib reduces IGF-1- stimulated HA secretion by OFs from TAO patients

Pretreatment with 10 μM linsitinib inhibit 100 ng/ml IGF-1-stimulated HA secretion from OFs [11]. We examined the effect of linsitinib on IGF-1-stimulated HA secretion from OFs. In our

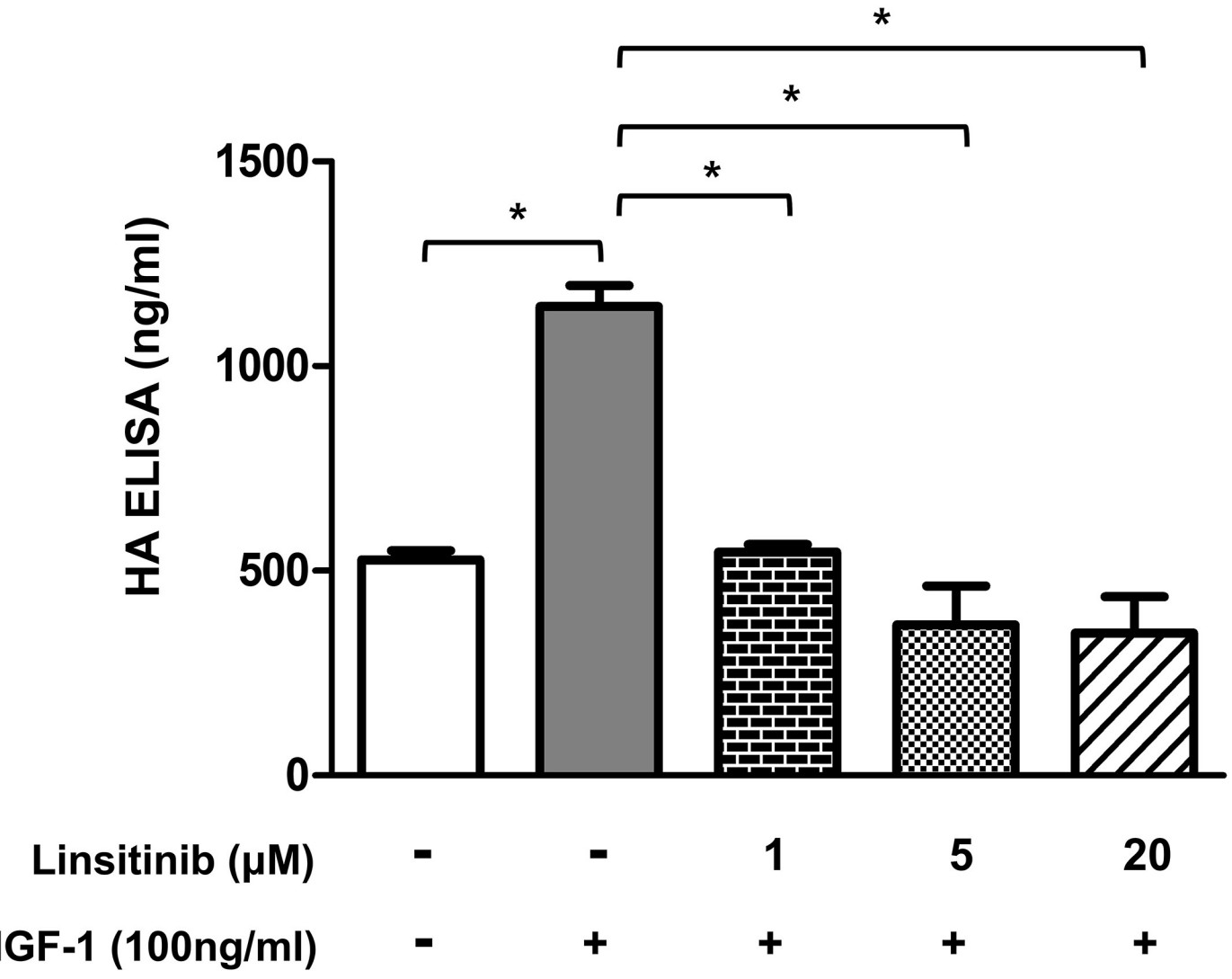

**Fig 3. Pretreatment with linsitinib reduces IGF-1- stimulated HA secretion by OFs from TAO patients. OFs from TAO patients were plated at $2 \times 10^5$ cells/ well in a 6-well plate.** After 24 h, cells were pre-treated with 1, 5 and 20 μM linsitinib for 2 h, followed by treatment with 100 ng/ml IGF-1 for 72 h. Culture medium was analyzed for HA concentration using a HA-enzyme linked immunosorbent assay (ELISA) kit (Corgenix, Westminster, USA) according to the manufacturer's instructions. The optical density for each well was then measured at 450 nm in a spectrometer (μ-Quant, BioTek instruments, Inc.). $^*P < 0.01$ between the indicated groups.

study, IGF-1 at a concentration of 50 ng/ml had relatively little effect on HA secretion from OFs, while IGF-1 at 100 ng/ml had a significant effect (data not shown). Therefore, although we observed the effect of linsitinib on cell proliferation in OFs treated with 50 ng/ml of IGF-1, the effect of linsitinib on HA secretion in OFs treated with 100 ng/ml of IGF-1. Consistent with the results of a previous study [11], our results showed that pretreatment with linsitinib significantly reduced 100 ng/ml IGF-1- stimulated HA secretion by OFs from TAO patients (Fig 3).

### Pretreatment with linsitinib reduces IGF-1-induced phosphorylation of IGF-1Rβ, Akt and ERK in OFs from TAO patients

We then examined the effect of linsitinib on the pathways of IGF-1Rβ, Akt, and ERK in IGF-1-treated OFs to investigate the mechanism underlying the effects of linsitinib on inhibiting

cell proliferation and HA secretion in OFs treated with IGF-1. Treatment with 50 ng/ml IGF-1 induced phosphorylation of IGF-1Rβ at Tyr1135 by 2.49 fold and Akt at Thr308 by 2.54 fold and Ser473 by 2.34 fold in OFs from TAO patients (Fig 4). Pretreatment with linsitinib, at 1, 5 and 20 μM inhibited IGF-induced phosphorylation of IGF-1Rβ at Tyr1135 and Akt at Ser473, but not at Thr308 of Akt, in OFs (Fig 4). In addition, treatment with 50 ng/ml IGF-1 induced phosphorylation of ERK by 2.18 fold (Fig 4). Pretreatment with 20, but not 1 and 5, μM linsitinib inhibited IGF-1-inuced phosphorylation of ERK (at 20 μM by 0.96 fold vs only IGF-1 treatment by 2.18 fold, Fig 4).

## Discussion

The findings reported herein indicate that pretreatment with linsitinib inhibited IGF-1-induced the phosphorylation of IGF-1Rβ, Akt at Ser473, and ERK, subsequently leading to the downregulation of the cell proliferation and HA secretion in the OFs of patients with TAO.

We initially investigated whether IGF-1 stimulates the proliferation of both OFs of patients with TAO and control subjects. However, IGF-1 stimulated only the proliferation of OFs from patients with TAO, not OFs from control subjects (Fig 1A and 1B). There may be several reasons why the effects of IGF-1 on cell proliferation in the OFs of patients with TAO differ from those in control subjects, but we thought that the difference in the expression levels of IGF-1R between the OFs of patients with TAO and control subjects might be involved. We previously reported that the OFs from TAO patients showed significantly high levels of IGF-1R expression on the cell surface, compared to OFs from control subjects [12].

Binding of IGF-1 or -2 to the α subunit of IGF-1R leads to the conformational change in the β subunit, resulting in the activation of receptor tyrosine kinase activity and the phosphorylation of IGF-1R, subsequently leading to the activation of downstream signalings, PI3K / Akt and ERK pathways [13,14].

Our results also show that treatment with IGF-1 induced proliferation (Fig 1A) and HA secretion (Fig 3), and the phosphorylation of IGF-1Rβ at Tyr1135, Akt at Thr308, and Ser473, and ERK (Fig 4) in OFs from TAO patients.

Linsitinib, known as OSI-906, is an orally efficacious dual inhibitor of IGF-1R and insulin receptor [8]. In the current study, we investigated whether linsitinib affects the signaling of IGF-1R in OFs of TAO patients. Our results showed that pretreatment with linsitinib inhibited IGF-1-induced cell proliferation (Fig 1A) and HA secretion (Fig 3) of OFs from TAO patients. Consistent with our results, Neumann et al. [15] reported that pretreatment with linsitinib inhibited IGF-1-induced HA secretion by OFs. In addition, Mulvihill et al. [8] also reported that pretreatment with linsitinib inhibited IGF-1-induced cell proliferation in various tumor cell lines. However, the results of the experiment performed by Mulvihill et al. [8] using mouse 3T3 cells overexpressing human IGF-1R to observe the effect of linsitinib on the Akt phosphorylation in the IGF-1R, PI3K/Akt, and ERK pathway differ from those of our experiment using the OFs of TAO patients.

Akt and ERK are components of downstream signaling of the IGF-1R pathway involved in TAO development [5]. The interaction of the Akt PH domain with phosphatidylinositol 3,4,5-trisphosphate (PIP3) is thought to induce conformational changes in Akt, resulting in exposure of its two main phosphorylation sites, Thr 308 and Ser 473. Full activation of Akt requires the phosphorylation of Thr 308 by PDK1 and Ser 473 by the mammalian target of rapamycin complex 2 (mTORC2) kinase, respectively [14,16,17]. However, the binding of growth factor receptor-bound 2 (GRB2) to IGF-1R substrates leads to the activation of

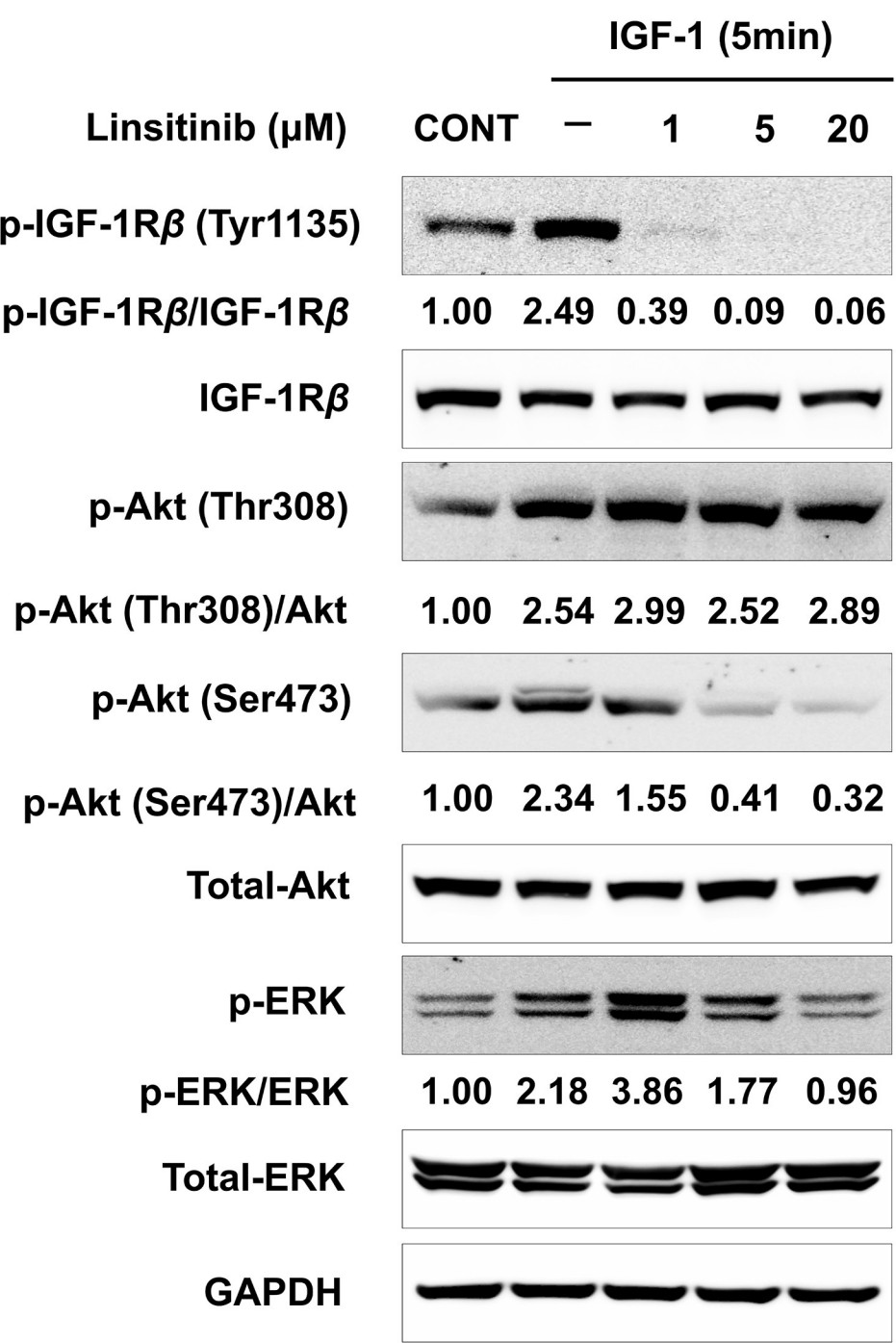

**Fig 4. Pretreatment with linsitinib reduces IGF-1-induced phosphorylation of IGF-1R, Akt and ERK in OFs from TAO patients. OFs from TAO patients were plated at a concentration of 2 x 10$^5$ cells/well in a 6-well plate.** After 24 h, cells were starved in serum-free medium for 24 h. The serum starved cells were pre-treated with linsitinib at the indicated concentrations for 2 h, followed by treatment with 50 ng/ml IGF-1 for 5 min. The levels of phosphorylation of IGF-1R, Akt and ERK were determined via western blot analysis. Similar results were observed in three independent experiments.

GRB2-associated SOS guanine nucleotide exchange activity, resulting in the activation of the Ras / Raf / MAPK (ERK) pathway [13].

The results obtained by Mulvihill et al. along with our results show that pretreatment with linsitinib inhibited IGF-1-induced phosphorylation of IGF-1R at Tyr1135, Akt at Ser473, and ERK (Fig 4). However, while the experimental results obtained by Mulvihill et al. [8] showed that pretreatment with linsitinb also inhibited IGF-1-induced phosphorylation of Akt at Thr308 in mouse 3T3 cells overexpressing human IGF-1R, our results show that pretreatment with linsitinb did not inhibit IGF-1-induced phosphorylation of Akt at Thr308 in the OFs of TAO patients (Fig 4).

Direct comparison of our results with those obtained by Mulvihill et al. [8] is difficult owing to differences in cell type (mouse 3T3 overexpressing human IGF-1R in the study by Mulvihill et al. vs human OFs of TAO patients in our study) and experimental conditions (phosphorylation level of Akt, 15 min after IGF-1 treatment in Mulvihill et al.'s study vs 5 min after IGF-1 treatment in ours).

Akt is a component of IGF-1R/PI3K signaling and linsitinib is an inhibitor of IGF-1R; however, the reason linsitinib inhibits the phosphorylation of Akt at Ser 473, which is phosphorylated by mTORC2 in response to IGF-1 stimulation, but fails to inhibit phosphorylation of Akt at 308, which is phosphorylated by PDK1. In addition, our results showed that only high concentration (20 μM) of linsitinib inhibited IGF-1- induced phosphorylation of ERK, whereas treatment with linsitinib at a relatively low concentration of 1 μM promoted IGF-1-induced phosphorylation of ERK (Fig 4). The reason why the effect of linsitinib on ERK phosphorylation varies depending on the concentration of linsitinib is not yet known. Therefore, addressing these issues will necessitate research on the interactions among IGF-1R/PI3K, Akt, PDK1, mTORC2, and ERK in the OFs of TAO patients.

The IGF-1R pathway has been an important target for anticancer drug development. IGF-1R inhibitor has been developed in three types: 1) monoclonal antibodies against IGF-1R, 2) monoclonal antibodies against IGF-1R ligands, 3) IGF-1R tyrosine kinase inhibitor. Linsitinib is one of the drugs developed as an IGF-1R tyrosine kinase inhibitor and has entered clinical trials for various cancers [18]. A phase 3 clinical trial reported that linsitinib treatment did not significantly increase overall survival in patients with adenocarcinoma [19]. However, the safety and effectiveness of linsitinib, either alone or as a combination anticancer treatment regimen, has been studied in patients with various cancers, such as multiple myeloma [20], gastrointestinal stromal tumor [21], and epithelial ovarian. In addition, linsitinib has been receiving attention not only for its anticancer effect, but also for its anti-inflammatory effects. Gulbins et. al. [10] reported that linsitinib inhibits the development and progression of TAO via the suppression of infiltration of T lymphocytes and macrophages in an experimental murine model for Graves' disease. Our results also support the possibility of linsitinib as a therapeutic agent for inflammatory diseases, especially TAO.

## Supporting information

**S1 Fig. Pretreatment with linsitinib reduces IGF-1-increased the proportion of S-phase cells in OFs from TAO patients.** Representative histogram of the gated cells in the G0/G1, S, and G2/M phases of linsitinib-pretreated or/and IGF-1-treated OFs from TAO patients. OFs from TAO patients were plated at a concentration of 2 x 10⁵ cells/well in a 6-well plate. After 24 h, the cells were pre-treated with linsitinib at the indicated concentrations for 2 h, followed by treatment with 50 ng/ml IGF-1 for 24 h. After 24 h, cells were collected and stained with propidium iodide (PI) and their DNA contents were analyzed by flow cytometry.
(PDF)

**S1 Raw images.**
(TIF)

**S2 Raw images.**
(TIF)

**S3 Raw images.**
(TIF)

**S4 Raw images.**
(TIF)

**S5 Raw images.**
(TIF)

**S6 Raw images.**
(TIF)

**S7 Raw images.**
(TIF)

**S8 Raw images.**
(TIF)

**S9 Raw images.**
(TIF)

**S10 Raw images.**
(TIF)

## Author Contributions

**Conceptualization:** Ji-Young Lee, Seong-Beom Lee.

**Investigation:** Ji-Young Lee.

**Methodology:** Seong-Beom Lee.

**Project administration:** Suk-Woo Yang, Ji-Sun Paik.

**Supervision:** Seong-Beom Lee, Suk-Woo Yang.

**Writing – review & editing:** Ji-Young Lee, Ji-Sun Paik.

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
