## [Decision Letter · Decision Letter 0]

18 Jun 2024

PONE-D-24-15829Linsitinib inhibits IGF-1-induced proliferation of orbital fibroblasts by suppressing PI3K/AKT and ERK pathwaysPLOS ONE

Dear Dr. Paik,

Thank you for submitting your manuscript to PLOS ONE. After careful consideration, we feel that it has merit but does not fully meet PLOS ONE’s publication criteria as it currently stands. Therefore, we invite you to submit a revised version of the manuscript that addresses the points raised during the review process.

**Thank you for submitting the following manuscript to PLOS ONE.**

**Please revise the manuscript according to the reviewers' comments and upload the revised file.**

We look forward to receiving your revised manuscript.

Kind regards,

Yung-Hsiang Chen, Ph.D.

Academic Editor

PLOS ONE

Additional Editor Comments:

Thank you for submitting the following manuscript to PLOS ONE.

Please revise the manuscript according to the reviewers' comments and upload the revised file.

Reviewers' comments:

Reviewer's Responses to Questions

**Comments to the Author**

1. Is the manuscript technically sound, and do the data support the conclusions?

Reviewer #1: Yes

Reviewer #2: Yes

Reviewer #3: No

Reviewer #4: Partly

2. Has the statistical analysis been performed appropriately and rigorously? 

Reviewer #1: Yes

Reviewer #2: Yes

Reviewer #3: No

Reviewer #4: No

3. Have the authors made all data underlying the findings in their manuscript fully available?

Reviewer #1: Yes

Reviewer #2: Yes

Reviewer #3: Yes

Reviewer #4: Yes

4. Is the manuscript presented in an intelligible fashion and written in standard English?

Reviewer #1: Yes

Reviewer #2: Yes

Reviewer #3: Yes

Reviewer #4: Yes

5. Review Comments to the Author

Reviewer #1: The article has an important topic for the medical filed. I suggest a few changes.

1. please write the abbreviation for mitogen activated kinase kinase-MAPKK line 82

2. exist studies on animals or clinical trials thar used linsitinib for TAO?

3. During discussion section, please deeply describe the connection between TAO -PI3K/AKT/mTOR and also MAPKK

4.

Reviewer #2: In the present study, authors have investigated the effect of linsitinib, IGF-1R inhibitor on orbital fibroblasts of TAO patients in vitro, on countering proliferation of cells. Linsitinib's suppression of orbital fibroblast proliferation stimulated by IGF-1 isn't entirely novel, but it does contribute to our understanding of its pharmacological effects. Linsitinib is known as a dual inhibitor of the insulin-like growth factor receptor (IGF-1R) and insulin receptor (IR). Since IGF-1 is a potent stimulator of cell proliferation, inhibiting its signaling pathway with linsitinib is a logical approach to controlling cell growth.

It is interesting to find 0.1% BSA actually enhanced stimulatory effect of IGF-1 in orbital fibroblasts, not FBS. I have some comments.

1. In introduction, the paragraph regarding IGF-1R is too lengthy and requires shortening.

2. I wonder why the experiments were peformed only in TAO cells, but not in normal controls.

3. Authors should indicate how many different cell samples from different individual was used for each experiment (figure legend)

4. In discussion"Comparing our findings with those of Mulvihill et al. [10] is challenging owing to differences in cell type and experimental conditions" - specifically what is the difference? It is not mentioned.

5. In discussion, "linsitinib has garnered attention not only for its anticancer effects but also for its anti-inflammatory effects." - specifically, what was the molecular change by linsitinib regarding inflammation?

6. Teprotumumab blocking IGF-1R blocked secretion of hyaluronan in previous report. Have you found the effect of Linsitinib with respect to hyaluronan production?

7. Several supplementary data can be eliminated.

Reviewer #3: Have the authors performed the same series of experiments in normal non-graves patient's orbital fibroblasts? Why or why not? What is there results of the control group?

What phase of activity or chronicity were the patients undergoing orbital decompression in? Were samples pooled? It is unclear how many patients contributed samples and whether or not these samples were pooled together?

Reviewer #4: The study by Lee et al. focuses on the effect of Linsitinib, a selective IGF-1R inhibitor, on orbital fibroblasts (OFs). The authors demonstrated that linsitinib suppressed IGF-1-induced phosphorylation of IGF-1Rβ, AKT, and ERK, reduced cell proliferation, cyclin D1 expression, and the proportion of S-phase cells in OFs from TAO patients. These findings indicate linsitinib's potential as a novel TAO treatment by targeting the PI3K/AKT and ERK pathways.

In this manuscript, the authors assessed the phosphorylation of IGF-1Rβ, Akt Ser473, Akt Thr308, and ERK after stimulation with IGF-1. They also evaluated the expression levels of Cyclin D1, cell proliferation, and the percentage of cells in the S phase. While their results suggest that linsitinib can inhibit OF proliferation, some findings are questionable. For example, linsitinib inhibited Akt phosphorylation at Ser473 but not at Thr308, and at low concentrations, it even increased ERK phosphorylation. The authors attribute these anomalies to the specific cell type used, suggesting that further studies are needed. However, this explanation is not entirely satisfactory from a scientific standpoint. I believe additional experiments are necessary to clarify these results and provide a more comprehensive understanding of the underlying mechanisms.

In the context of TAO, activated orbital fibroblasts (OFs) not only increase their proliferation but also produce inflammatory mediators, differentiate into adipocytes and myofibroblasts, and produce excess amounts of extracellular matrix components such as hyaluronan. To substantiate their claim that linsitinib can be used as a treatment for TAO, the authors should examine at least some of these features in their experiments. This would provide a more comprehensive understanding of linsitinib's therapeutic potential and its effects on the various pathological processes associated with TAO.

The authors analyzed the significance of their data using Student's t-test for comparisons between two points. However, they did not specify the type of analysis employed for comparisons involving multiple data points. To ensure the robustness and reliability of their findings, it is important for the authors to clarify and include appropriate statistical analyses for multiple comparisons, such as ANOVA followed by post hoc tests, in their manuscript.

6. PLOS authors have the option to publish the peer review history of their article (what does this mean?). If published, this will include your full peer review and any attached files.

Reviewer #1: No

Reviewer #2: No

Reviewer #3: No

Reviewer #4: No

---

## [Author Response · Author response to Decision Letter 0]

3 Sep 2024

Our detailed responses to the reviewers’ comments are listed below. 

Reviewer #1: 

The article has an important topic for the medical filed. I suggest a few changes.

1. please write the abbreviation for mitogen activated kinase kinase-MAPKK line 82

: As the reviewer suggested, we used the abbreviation for mitogen activated kinase kinase-MAPKK in the revised manuscript. 

2. exist studies on animals or clinical trials that used linsitinib for TAO?

: There is an animal model of TAO disease and we cited the paper regarding the effect of linsitinib in that animal model (Frontiers in Endocrinology 14: 1211473, 2023).

3. During discussion section, please deeply describe the connection between TAO -PI3K/AKT/mTOR and also MAPKK

: As the reviewer suggested, we described ‘the connection between TAO-PI3K/AKT/mTOR and also MAPKK’ in more detail, in the section of discussion in the revised manuscript, as described below.

‘Akt and ERK are components of downstream signaling of IGF-1R pathway known to be involved in TAO development. The interaction of Akt PH domain with phosphatidylinositol 3,4,5-trisphosphate (PIP3) is thought to induce conformational changes in Akt, resulting in exposure of its two main phosphorylation sites, Thr 308 and Ser 473. For full activation of Akt, it needs phosphorylation of Thr 308 by PDK1 and Ser 473 by mammalian target of rapamycin complex 2 (mTORC2) kinase, respectively [13-15]. On the other hand, the binding of growth factor receptor-bound 2 (GRB2) to IGF-1R substrates leads to the activation of GRB2-associated SOS guanine nucleotide exchange activity, resulting activation of Ras / Raf / MAPK (ERK) pathway [9].’ 

Reviewer #2: 

In the present study, authors have investigated the effect of linsitinib, IGF-1R inhibitor on orbital fibroblasts of TAO patients in vitro, on countering proliferation of cells. Linsitinib's suppression of orbital fibroblast proliferation stimulated by IGF-1 isn't entirely novel, but it does contribute to our understanding of its pharmacological effects. Linsitinib is known as a dual inhibitor of the insulin-like growth factor receptor (IGF-1R) and insulin receptor (IR). Since IGF-1 is a potent stimulator of cell proliferation, inhibiting its signaling pathway with linsitinib is a logical approach to controlling cell growth.

It is interesting to find 0.1% BSA actually enhanced stimulatory effect of IGF-1 in orbital fibroblasts, not FBS. I have some comments.

1. In introduction, the paragraph regarding IGF-1R is too lengthy and requires shortening.

: As the reviewer suggested, we reduced the length of the paragraph related to IGF-1R in the revised version. 

2. I wonder why the experiments were performed only in TAO cells, but not in normal controls.

: As the reviewer suggested, we examined the effect of IGF-1 on the proliferation of OFs from patients with TAO and control subjects, and prepared new figure (Fig.1B). However, treatment with 50 ng/mL IGF-1 did not increase the proliferation of OFs from control subjects (Fig.1B). Therefore, in this study, we studied the effect of linstinib only on OFs from TAO patients. 

In addition, we described the reasons why the effects of IGF-1 on cell proliferation were different in OFs from patients with TAO and control subjects in the section of discussion, as described below. .

‘We initially investigated the issue of whether IGF-1 stimulates the proliferation of both OFs of patients with TAO and control subjects. However, IGF-1 stimulated only the proliferation of OFs from patients with TAO, not OFs from control subjects (Fig. 1A and B). There may be several reasons why the effects of IGF-1 on cell proliferation are different in OFs of patients with TAO and control subjects, but we thought that the difference in the expression levels of IGF-1R between OFs of patients with TAO and control subjects might be involved. We previously reported that the OFs from TAO patients showed significantly higher levels of IGF-1R expression on the cell surface of the OFs from TAO, compared to OFs from control subjects [8].’

3. Authors should indicate how many different cell samples from different individual was used for each experiment (figure legend)

: The editors of Plos One asked us for complete anonymity of each experiment, we could not indicate the cell numbers in the figure legend. However, the reviewer suggested, we added patients’ information that obtained OFs with/without TAO and summarized in Table 1.

4. In discussion "Comparing our findings with those of Mulvihill et al. [10] is challenging owing to differences in cell type and experimental conditions" - specifically what is the difference? It is not mentioned.

: As the reviewer suggested, we described the differences between Mulvihill et al.’s and our experimental results in more detail in the revised manuscript. 

‘Both Mulvihill MJ et al.’s and our results show that pretreatment with linsitinib inhibited IGF-1-induced phosphorylation of IGF-1R at Tyr1135, Akt at Ser473, and ERK1/2 (Fig. 4). However, while the experimental results of Mulvihill MJ et al. [10] showed that pretreatment with linsitinb also inhibited IGF-1-induced phosphorylation of Akt at Thr308, our results show that pretreatment with linsitinb did not inhibit IGF-1-induced phosphorylation of Akt at Thr308 (Fig. 4).

It is difficult to directly compare our results with those of Mulvihill MJ et al. [10] study due to differences in cell type (mouse 3T3 overexpressing human IGF-1R in Mulvihill MJ et al.’s vs human OFs of TAO patients in ours) and experimental conditions (phosphorylation level of Akt, 15 min after IGF-1 treatment in Mulvihill MJ et al.’s vs 5 min after IGF-1 treatment in ours). 

Akt is a component of IGF-1R/PI3K signaling and linsitinib is an inhibitor of IGF-1R, but it is not yet clear why, in OFs, linsitinib inhibits phosphorylation of Akt at Ser 473, which is phosphorylated by mTORC2 in response to IGF-1 stimulation, but fails to inhibit phosphorylation of Akt at 308, which is phosphorylated by PDK1. In addition, our results showed that only high concentration (20 μM) of linsitinib inhibited IGF-1- induced phosphorylation of ERK, whereas treatment with linsitinib at a relatively low concentration of 1 μM promoted IGF-1-induced phosphorylation of ERK (Fig. 4). The reason why the effect of linsitinib on ERK phosphorylation varies depending on the concentration of linsitinib is not yet known. Thus, to address these issues, it will be necessary to study the interactions among IGF-1R/PI3K, Akt, PDK1, mTORC2, and ERK in OFs of TAO patients.’

5. In discussion, "linsitinib has garnered attention not only for its anticancer effects but also for its anti-inflammatory effects." - specifically, what was the molecular change by linsitinib regarding inflammation?

: Gulbin A et al. (Frontiers in Endocrinology 14:1211473, 2023) reported that treatment with linsitinib attenuates the development and progression of TAO by inhibiting the infiltration of T lymphocytes and macrophages. Therefore, we thought that linsitinib also has an anti-inflammatory effect.

6. Teprotumumab blocking IGF-1R blocked secretion of hyaluronan in previous report. Have you found the effect of Linsitinib with respect to hyaluronan production?

: As the reviewer suggested, we investigated the issue of whether linsitinib inhibits IGF-1-stimulated HA secretion in OFs of TAO patients and made a new Figure (Figure 3) showing the results of linsitinib's inhibition of HA secretion induced by IGF-1 stimulation, as described below. In addition, we changed the title of this article from ‘Linsitinib inhibits IGF-1-induced proliferation of orbital fibroblasts through suppressing PI3K/AKT and ERK pathway’ to ‘Linsitinib inhibits IGF-1-induced cell proliferation and hyaluronic acid secretion through suppressing PI3K/Akt and ERK pathway in orbital fibroblasts from patients with thyroid-associated ophthalmopathy.’ 

‘We also examined the effect of linsitinib on IGF-1-stimulated HA secretion from OFs. In our study, IGF-1 at a concentration of 50 ng/ml had a small effect on HA secretion from OFs, while IGF-1 at 100 ng/ml had a significant effect (data not shown). Therefore, although we observed the effect of linsitinib on cell proliferation in OFs treated with 50 ng/ml of IGF-1, the effect of linsitinib on HA secretion in OFs treated with 100 ng/ml of IGF-1.

Consistent with the previous result (#6. Endocrinology. 2017;158 (10):3676-83), our results also showed that pretreatment with linsitinib significantly reduced 100 ng/ml IGF-1- stimulated HA secretion by OFs from TAO patients (Fig. 3).’

7. Several supplementary data can be eliminated.

Reviewer #3: 

Have the authors performed the same series of experiments in normal non-graves patient's orbital fibroblasts? Why or why not? What is there results of the control group?

: As the reviewer suggested, we examined the effect of IGF-1 on the proliferation of OFs from patients with TAO and control subjects, and added a new figure (Fig.1B). However, treatment with 50 ng/mL IGF-1 did not increase the proliferation of OFs from control subjects (Fig.1B). Therefore, in this study, we studied the effect of linstinib only on OFs from TAO patients. 

In addition, we described the reasons why the effects of IGF-1 on cell proliferation were different in OFs from patients with TAO and control subjects in the section of discussion, as described below.

‘We initially investigated the issue of whether IGF-1 stimulates the proliferation of both OFs of patients with TAO and control subjects. However, IGF-1 stimulated only the proliferation of OFs from patients with TAO cells, not OFs from control subjects (Fig. 1A and B). There may be several reasons why the effects of IGF-1 on cell proliferation are different in OFs of patients with TAO and control subjects, but we thought that the difference in the expression levels of IGF-1R between OFs of patients with TAO and control subjects might be involved. We previously reported that the OFs from TAO patients showed significantly higher levels of IGF-1R expression on the cell surface of the OFs from TAO, compared to OFs from control subjects [8].’

What phase of activity or chronicity were the patients undergoing orbital decompression in? Were samples pooled? It is unclear how many patients contributed samples and whether or not these samples were pooled together?

: In general, decompression surgery was perfomed in the chronic stage, so the cells acquired for this research were also same condition as chronic phase. In addition, emergency surgery due to optic neuropathy was not included in this experiment. As the reviewer suggested, OFs related informations pariticipating in this experiment were added to the manuscripts.

Reviewer #4: 

The study by Lee et al. focuses on the effect of Linsitinib, a selective IGF-1R inhibitor, on orbital fibroblasts (OFs). The authors demonstrated that linsitinib suppressed IGF-1-induced phosphorylation of IGF-1Rβ, AKT, and ERK, reduced cell proliferation, cyclin D1 expression, and the proportion of S-phase cells in OFs from TAO patients. These findings indicate linsitinib's potential as a novel TAO treatment by targeting the PI3K/AKT and ERK pathways.

In this manuscript, the authors assessed the phosphorylation of IGF-1Rβ, Akt Ser473, Akt Thr308, and ERK after stimulation with IGF-1. They also evaluated the expression levels of Cyclin D1, cell proliferation, and the percentage of cells in the S phase. While their results suggest that linsitinib can inhibit OF proliferation, some findings are questionable. For example, linsitinib inhibited Akt phosphorylation at Ser473 but not at Thr308, and at low concentrations, it even increased ERK phosphorylation. The authors attribute these anomalies to the specific cell type used, suggesting that further studies are needed. However, this explanation is not entirely satisfactory from a scientific standpoint. I believe additional experiments are necessary to clarify these results and provide a more comprehensive understanding of the underlying mechanisms.

: Although we agree with the reviewer’s opinion, however, within the allocated time frame, it was not possible to clarify the reason why linsitinib inhibits phosphorylation of Akt at Ser 473, which is phosphorylated by mTORC2 in response to IGF-1 stimulation, but fails to inhibit phosphorylation of Akt at 308, which is phosphorylated by PDK1 in OFs of TAO patients. So we described the limitation of our study as described below. 

‘Akt is a component of IGF-1R/PI3K signaling and linsitinib is an inhibitor of IGF-1R, , but it is not yet clear why, in OFs, linsitinib inhibits phosphorylation of Akt at Ser 473, which is phosphorylated by mTORC2 in response to IGF-1 stimulation, but fails to inhibit phosphorylation of Akt at 308, which is phosphorylated by PDK1. In addition, our results showed that only high concentration (20 μM) of linsitinib inhibited IGF-1- induced phosphorylation of ERK, whereas treatment with linsitinib at a relatively low concentration of 1 μM promoted IGF-1-induced phosphorylation of ERK (Fig. 4). The reason why the effect of linsitinib on ERK phosphorylation varies depending on the concentration of linsitinib is not yet known. Thus, to address these issues, it will be necessary to study the interactions among IGF-1R/PI3K, Akt, PDK1, mTORC2, and ERK1/2 in OFs of TAO patients.’

In the context of TAO, activated orbital fibroblasts (OFs) not only increase their proliferation but also produce inflammatory mediators, differentiate into adipocytes and myofibroblasts, and produce excess amounts of extracellular matrix components such as hyaluronan. To substantiate their claim that linsitinib can be used as a treatment for TAO, the authors should examine at least some of these features in their experiments. This would provide a more comprehensive understanding of linsitinib's therapeutic potential and its effects on the various pathological processes associated with TAO.

: As the reviewer suggested, we investigated the issue of whether linsitinib inhibits IGF-1-stimulated HA secretion in OFs of TAO patients and made a new Figure (Figure 3) showing the results of linsitinib's inhibition of HA secretion induced by IGF-1 stimulation, as described below. 

In addition, we changed the title of this article from ‘Linsitinib inhibits IGF-1-induced proliferation of orbital fibroblasts through suppressing PI3K/AKT and ERK pathway’ to ‘Linsitinib inhibits IGF-1-induced cell proliferation and hyaluronic acid secretion through suppressing PI3K/Akt and ERK pathway in orbital fibroblasts from patients with thyroid-associated ophthalmopathy.’ 

‘We also examined the effect of linsitinib on IGF-1-stimulated HA secretion from OFs. In our study, IGF-1 at a concentration of 50 ng/ml had a small effect on HA secretion from OFs, while IGF-1 at 100 ng/ml had a significant effect (data not shown). Therefore, although we observed the effect of linsitinib on cell proliferation in OFs treated with 50 ng/ml of IGF-1, the effect of linsitinib on HA secretion in OFs treated with 100 ng/ml of IGF-1.

Consistent with the previous result (#6. Endocrinology. 2017;158 (10):3676-83), our results also showed that pretreatment with linsitinib significantly reduced 100 ng/ml IGF-1- stimulated HA secretion by OFs from TAO patients (Fig. 3).’

The authors analyzed the significance of their data using Student's t-test for comparisons between two points. However, they did not specify the type of analysis employed for comparisons involving multiple data points. To ensure the robustness and reliability of their findings, it is important for the authors to clarify and include appropriate statistical analyses for multiple comparisons, such as ANOVA followed by post hoc tests, in their manuscript.

: As the reviewer suggested, we used the one-way analysis of variance (ANOVA) for statistical analysis in the revised manuscript.

---

## [Decision Letter · Decision Letter 1]

13 Sep 2024

Linsitinib inhibits IGF-1-induced cell proliferation and hyaluronic acid secretion by suppressing PI3K/Akt and ERK pathway in orbtial fibroblasts from patients with thyroid-associated ophthalmopathy

PONE-D-24-15829R1

Dear Dr. Paik,

We’re pleased to inform you that your manuscript has been judged scientifically suitable for publication and will be formally accepted for publication once it meets all outstanding technical requirements.

Kind regards,

Yung-Hsiang Chen, Ph.D.

Academic Editor

PLOS ONE

Additional Editor Comments (optional):

Congratulations on the acceptance of your manuscript, and thank you for your interest in submitting your work to PLOS ONE.

Reviewers' comments:

Reviewer's Responses to Questions

**Comments to the Author**

1. If the authors have adequately addressed your comments raised in a previous round of review and you feel that this manuscript is now acceptable for publication, you may indicate that here to bypass the “Comments to the Author” section, enter your conflict of interest statement in the “Confidential to Editor” section, and submit your "Accept" recommendation.

Reviewer #1: All comments have been addressed

Reviewer #2: All comments have been addressed

2. Is the manuscript technically sound, and do the data support the conclusions?

Reviewer #1: Yes

Reviewer #2: Yes

3. Has the statistical analysis been performed appropriately and rigorously? 

Reviewer #1: Yes

Reviewer #2: Yes

4. Have the authors made all data underlying the findings in their manuscript fully available?

Reviewer #1: Yes

Reviewer #2: Yes

5. Is the manuscript presented in an intelligible fashion and written in standard English?

Reviewer #1: Yes

Reviewer #2: Yes

6. Review Comments to the Author

Reviewer #1: I do not have any remark for the authors. Authors improved their article according to reviewers comments

Reviewer #2: Authors have responded thoroughly to the reviewer's comments. The quality of paper has improved after the addition of HA result. I believe the reviewer version is acceptable. I have no further comments.

7. PLOS authors have the option to publish the peer review history of their article (what does this mean?). If published, this will include your full peer review and any attached files.

Reviewer #1: No

Reviewer #2: **Yes: **Jin Sook Yoon

---

## [Editor Report · Acceptance letter]

9 Oct 2024

PONE-D-24-15829R1 

PLOS ONE

Dear Dr. Paik, 

I'm pleased to inform you that your manuscript has been deemed suitable for publication in PLOS ONE. Congratulations! Your manuscript is now being handed over to our production team.

Kind regards, 

on behalf of

Dr. Yung-Hsiang Chen 

Academic Editor

PLOS ONE